# The Transabdominal Lumbar Approach (TALA) for Robotic Renal Surgery—A Retrospective Single-Center Comparative Study and Step-by-Step Description of a Novel Approach

**DOI:** 10.3390/cancers16020446

**Published:** 2024-01-20

**Authors:** Franziska Maria Heining, Uwe Bieri, Tilo Niemann, Philipp Maletzki, Christopher Tschung, Jean-Pascal Adank, Fabian Rössler, Antonio Nocito, Lukas John Hefermehl

**Affiliations:** 1Department of Surgery, Division of Urology, Kantonsspital Baden, 5404 Baden, Switzerland; franziska-heining@gmx.de (F.M.H.); uwe.bieri@ksb.ch (U.B.); philipp.maletzki@ksb.ch (P.M.); jean-pascal.adank@ksb.ch (J.-P.A.); 2Department of Radiology, Kantonsspital Baden, 5404 Baden, Switzerland; tilo.niemann@ksb.ch; 3Department of Surgery and Transplantation, University Hospital Zürich, University of Zürich, 8006 Zürich, Switzerland; fabian.roessler@usz.ch; 4Department of Surgery, Kantonsspital Baden, 5404 Baden, Switzerland; antonio.nocito@ksb.ch

**Keywords:** robotic surgery, partial nephrectomy, nephrectomy, surgical technique, laparoscopy

## Abstract

**Simple Summary:**

Robotic kidney cancer surgery is commonly performed by two different approaches: the transperitoneal approach (TP) and the retroperitoneal approach (RP). Both methods have challenges, such as limited space (RP) or difficulties in dissecting the renal artery (TP). A combination of both methods, called a hybrid approach, has been described before but not fully evaluated. The study proposed a new modified hybrid approach called the transabdominal lumbar approach (TALA). The study compared 20 consecutive patients undergoing RP and 20 patients using TALA. The study looked at factors such as operation time, blood loss, and complications. The study found that both methods were similar in most areas. In conclusion, TALA is a safe and promising approach that combines the benefits of RP and TP for kidney cancer surgery.

**Abstract:**

The transperitoneal approach (TP) and the retroperitoneal approach (RP) are two common methods for performing nephrectomy or partial nephrectomy. However, both approaches face difficulties, such as trocar placement and limited working space (RP). TP is impaired in the case of dorsal tumors and dissection of the renal artery can be challenging due to the anatomic localization dorsally to the renal vein. A hybrid approach that combines both methods has been previously reported in a case series, but not evaluated systematically. This study proposes a modified hybrid approach, which we call the transabdominal lumbar approach (TALA), involving late robotic docking after elaborating the retroperitoneum using conventional laparoscopy. The study compares the last 20 consecutive patients who underwent RP and the last 20 patients who underwent TALA at our institution. The investigated variables include operative time and amount of blood loss, hospitalization duration, postoperative analgesia requirement, and postoperative complications. The study found no significant difference in operative time, blood loss, ischemia time, or hospital stay between the two groups. The TALA group had fewer complications regarding Clavien–Dindo category 3, but one complication of category 4. In Conclusion, TALA is a safe and promising approach that combines the advantages of RP and TP.

## 1. Introduction

Minimal-invasive laparoscopic surgery has become a standard approach for total (TN) and partial nephrectomy (PN) in treating renal cancer. Compared to open surgery, the laparoscopic approach has several advantages, including lower complication rates, less blood loss, shorter hospitalization time, and more tumor-free (R0) resection margins [1,2]. Two different surgical approaches are commonly used to perform laparoscopic or robotic TN and PN: the transperitoneal approach (TP) and the retroperitoneal approach (RP). When performing the TP approach, all trocars are placed transabdominally following an oblique line between the ipsilateral lower lateral abdomen and the xiphoid process. This approach has the advantage of easy trocar placement access and a larger working space with minimal instrument conflicts. When surgical situs is accessed via the RP approach, then the trocars are placed in a manner to directly access the retroperitoneal cavity following the lumbar line (dorsally between the iliac crest and the lowest rib and ventrally in the direction of the umbilicus). The artificial cavity must be established with pressure applied by a balloon and through conventional laparoscopy before the remaining robotic trocars can be introduced. This technique has the advantage of gaining direct access to the renal artery by entering dorsally to the kidney due to the trocars lined up in the lumbar line. 

A meta-analysis comparing 21 studies on the differences between these two laparoscopic approaches showed a significantly shorter operation time and hospitalization time and less blood loss in the RP group compared to the TP group. Additionally, no significant difference in terms of perioperative complications and resection margins was noted [3]. Other advantages of the RP approach include simplified access to the hilar structures, allowing easier and faster dissection and control of the hilar vessels. In addition, the retroperitoneum can act as a tamponade space in case of postoperative bleeding and reduces the risk of peritonitis in case of urinary fistulation or leakage. However, the RP approach is associated with a shallower learning curve compared to the TP approach [4]. Therefore, RP surgery is generally considered to be technically more challenging for three reasons: (1) difficulty in identifying landmarks with risk of injury to vascular structures, (2) difficulty in creating the workspace and (3) operating in the limited working space of the retroperitoneum. Thus, most surgeons still prefer the TP approach [5]. However, the TP approach also poses some inherent challenges to the surgeon. By entering from a ventral angle, the renal vein is then usually located directly in front of the renal artery, which complicates and adds an additional degree of difficulty in achieving control over the renal vessels (vein and artery) compared to the RP approach [6]. Considering these critical points, the logical subsequent step seems to modify the access in such a way that the advantages of both approaches are combined and at the same time the described disadvantages are eliminated.

A hybrid combination approach that combines the advantages of the RP and the TP approach has been described in a video case report and was called the “trans-retro approach” by Regmi and coworkers [7]. However, they reported a single case of a complicated PN with an operation time of over 5 h. To our knowledge, no other reports of a similar approach have been published yet. 

At our department, we have established a technique of such a hybrid approach, aiming to continuously improve the surgical technique and patient outcome: the transabdominal lumbar approach (TALA). We believe that, especially considering the specific problems of RP and TP access discussed, modified access is needed and that the TALA approach offers the possibility of combining the advantages of both worlds. 

## 2. Materials and Methods

### 2.1. Patient Selection

In this retrospective study, we compared all consecutive TALA cases compared to the same number of the last consecutive RP cases performed in our hospital until we reached the same number of patients in both groups. After TALA was introduced in our institution, it was generally used more often. The decision for access for a specific patient ultimately remained a team decision. The study was conducted in accordance with the Declaration of Helsinki, and approved by the Cantonal Ethics Committee (Basec-No. 2023-00264). All patients signed a document for the use of data and images.

### 2.2. Investigated Variables

The evaluated variables of the study were duration of surgery (minutes), blood loss (mL) including a comparison of pre-and postoperative hemoglobin (g/dL) and creatinine levels (µmol/L), duration of hospital stay (days) and postoperative analgesic requirements (based on WHO-Pain Relief Classification (level 1–level 3) [8]). We also reported the duration of ischemia time during surgery for PN (minutes). Postoperative complications were classified by the validated Clavien–Dindo classification [9]), with readmission and reoperation rates. Previous abdominal surgeries in medical history were recorded. To classify the patient-specific comorbidities, we used the American Society of Anesthesiologists (ASA) score [10]. 

### 2.3. Statistics

Continuous normally and non-normally distributed variables were reported as the median and range. Categorical variables were reported as counts and percentages. The Mann–Whitney U test and Fisher’s exact probability test were used to evaluate group differences between the TP and TALA approaches. All data analyses were performed using SPSS statistical software Version 28 (IBM SPSS, Chicago, IL, USA). Statistical significance was considered when *p* was less than or equal to 0.05 in all analyses.

### 2.4. Surgical Technique 

#### 2.4.1. The Retroperitoneal Approach (RP)

Patients were positioned in a bent 70° flank position. All trocars are placed in the retroperitoneum. The first incision is performed in the axillar line near the tip of the 12th rib. After blunt incision through fascia and muscle the large pean clamp is opened to dilate the canal. Next, a balloon trocar is placed and inflated under visual control by the inserted camera. After positioning the second trocar lateral to the first in the lumbar line (dorsally between the iliac crest and the lowest rib and ventrally in the direction of the umbilicus), the peritoneum is gently pushed away with a laparoscopic scissor. Robotic trocars are placed medially in the lubar line before a 12 mm AirSeal trocar is placed between the two last trocars, followed by the docking of the robot. We used a Xi da Vinci robotic system (Intuitive, Sunnyvale, CA, USA) for all cases. In the first step, a dorso-lateral incision of the Gerota’s fascia was performed. The surface of the psoas muscle was identified. The nearly avascular plain was followed by holding up the kidney with a grasper. Next, the renal artery was identified. In the case of nephrectomy, the artery and later the renal vein were clipped using three Hem-o-lock (Teleflex, Morrisville, NC, USA) clips and cut. The kidney was then freed from the surrounding tissue after clipping and cutting the ureter. In the case of a partial nephrectomy, after identifying the renal artery, the tumor area was identified and the kidney surface was freed from surrounding fatty tissue. The borders were marked using an intraoperative ultrasound device (Hitachi, Tokyo, Japan). After clamping of the renal artery, the tumor was cut out from the surrounding healthy tissue and put into a retrieval bag. After inner renorrhaphy with V-lock 3-0 (Medtronic, Minneapolis, MN, USA), clamping was terminated. The surgery was completed after renorrhapy of the renal parenchyma using a running V-lock 3-0 suture, which was reinforced with Hem-o-lok Clips after each stitch.

#### 2.4.2. The Transabdominal Lumbar Approach (TALA)

Like the RP approach, the trocars are placed in the lumbar line after the patient is positioned in a bent 70° flank position. Figure 1 provides a step-by-step instruction for the correct trocar placement (Figure 1). Contrary to the RP approach, the first trocars are inserted transabdominally. The first trocar is placed at the intersection of the lumbar line and the lateral border of the rectus abdominis muscle via mini laparotomy access. This trocar marks the medial margin of trocar placement. After installation of the pneumo-peritoneum, two additional DaVinci trocars are placed under direct vision. One is inserted at a distance of about 8 cm laterally on the lumbar line. The other is slightly triangulated about 4 cm caudal to the lumbar line between the other two trocars. The camera is guided through the most medial trocar by the assistant surgeon. The surgeon then introduces laparoscopic scissors and a grasper over the two medial trocars. The retroperitoneum is entered after transection at the level of Toldt’s line (Figure 1B). The retroperitoneal fat is removed from the lateral abdominal wall, partly bluntly and partly with sharp dissection. This is followed by the placement of the remaining two trocars on the lumbar line, with a distance of 6–8 cm in between. Finally, the trocar previously inserted caudal to the lumbar line is removed and replaced with a 12 mm AirSeal trocar (Figure 1D). Initially, the slightly smaller trocar simplifies the conventional laparoscopic procedure due to the better support of the instruments. We used a constant pressure of 12 mmHg in all cases.

The robot is then placed in a standard renal surgery position behind the back of the patient. The DaVinci^®^ Instruments are then docked according to the standard protocol (Figure 2A). Figure 2 provides a detailed description of the most important steps of a common procedure. Additionally, the procedure follows the general principles of a standard laparoscopic nephrectomy or partial nephrectomy. 

## 3. Results

### 3.1. Patient Characteristics and Tumor Characteristics

Between June 2020 and October 2022, a total of 40 consecutive DaVinci-assisted TNs and PNs were performed at Cantonal Hospital Baden. The cohort consisted of 14 female and 26 male patients, with an average age of 63.5 years (27–84 years). The Body–Mass Index varied between 17.8 and 35.9. The standard RP approach was performed in 20 patients and the modified TALA approach in 20 patients. Twenty-seven of 40 (67.5%) patients were treated for a malignant renal tumor, 10 (25%) for a benign renal tumor, and three (7.5%) because of complications due to a nonfunctioning kidney. In total, 16 TNs and 24 PNs were performed. All surgeries were performed by the same experienced robotic surgeon and a surgeon in robotic training.

The RP group consisted of 8 women (40%) and twelve men (60%). Seven were scheduled for TN and 13 for PN. Nine patients had undergone abdominal surgery previously. The median age of the patients at the time of surgery was 65 years (27–81 years), and the median BMI was 28.3 kg/m^2^ (22.1–34.3 kg/m^2^). 

The TALA group consisted of 6 women (30%) and 14 men (70%). Nine were scheduled for TN and 11 for PN. Five patients had undergone abdominal surgery previously. The median age of the patients at the time of surgery was 67 years (41–84 years), and the median BMI was 27.5 kg/m^2^ (17.8–35.9 kg/m^2^). 

Considering tumor characteristics, histologic examination revealed benign findings in seven patients in the RP group and in three patients in the TALA group. Malignant findings were seen in 12 patients in the RP group and in 15 patients in the TALA group, with clear cell renal cell carcinoma being the most common in both groups (nine patients in the RP group and 13 patients in the TALA group). TN due to nonfunctioning kidneys was performed in one patient in the RP group and in two patients in the TALA group. Patient and tumor characteristics are summarized in Table 1.

### 3.2. Intraoperative and Postoperative Characteristics

The median operation time in the RP group was 211 min (154–289 min), in the TALA group 207 min (124–306 min) (*p* = 0.38). Median blood loss was 175 mL (50–450 mL) in the RP group and 175 mL (30–800 mL) in the TALA group (*p* = 0.8). Related to the partial nephrectomies, the median warm ischemia time was 15 min (9–30 min) in the RP group and 17 min (11–22 min) in the TALA group (*p* = 0.76) (Table 2).

Median hospitalization time was 6 days (4–9 days) in the RP group and 6 days (4–8 days) in the TALA group (*p* = 0.80). The postoperative analgesic requirements in the first five postoperative days, based on WHO classification, were as follows: On the day of surgery, patients in the TALA group and the RP group required median WHO level three analgesics (*p* = 1.0). On postoperative day (POD) one, patients in the RP group required WHO level two analgesics, and patients in the TALA group required WHO level one analgesics (*p* = 0.53). On PODs two, three, four, and five, both groups required WHO level one analgesics (*p* = 0.12, 0.16, 0.59, 0.29). Comparing median preoperative to postoperative hemoglobin (Hb) levels, a decrease of 2.5 g/dL in the TALA group and 2.4 g/dL in the RP group (*p* = 0.60) was noted. The median change in creatinine level from preoperative to postoperative was 29 µmol/L in the TALA group and 8 µmol/L in the RP group (*p* = 0.035) (Table 2).

### 3.3. 30-Day Morbidity

In the RP group, three Clavien–Dindo category III complications were documented. Two patients developed a pneumothorax, and one patient developed a trocar-incision hernia. In the TALA group two complications of category three occurred; one patient developed a superinfection of a postoperative hematoma in the renal fossa requiring drainage insertion and another patient developed an upper gastrointestinal tract hemorrhage that required transfusion of four red blood cell concentrates and one complication of category four, a duodenal ulceration four weeks after surgery that required surgical management. In the TALA group, three patients received red blood cell transfusions compared to no patient in the RP group. Considering 30-day morbidity, one patient in the RP group was admitted to IMC (intermediate care unit) postoperatively due to hyponatremia. In the TALA group, a total of five patients were rehospitalized within 30 days: One patient due to duodenal perforation admitted to ICU (intensive care unit), one patient due to upper gastrointestinal bleeding admitted to IMC (intermediate care unit), and three patients admitted to regular wards (one patient due to status epilepticus most likely in the setting of pneumonia, one patient due to infected hematoma, and one patient for general condition deterioration) (Table 2).

## 4. Discussion

This is, to the best of our knowledge, the first study to examine the role of the transabdominal lumbar approach technique for minimally invasive nephrectomy, in comparison with the retroperitoneal approach. Special regard was held for postoperative complications and possible technical improvements. Our single-center data confirm safety and feasibility of this new hybrid technique. This investigation serves as a pilot study in preparation for a planned randomized trial. Even though we only used the da Vinci system in this study, we assume that the principle could also be adapted for other robotic platforms.

Our study comparing TALA and RP revealed no difference in hospital stay (6 days) and ischemia time (15 min vs. 17 min) when PN was performed. In terms of operation time and blood loss, no significant difference was shown in both groups, with a trend in the TALA group toward shorter operation time.

Analysis of the comparison of preoperative and postoperative hemoglobin level changes between the two groups revealed no significant difference. Only the increase in the postoperative creatinine value compared to the preoperative value showed a significantly higher increase in the TALA group compared to the RP group (*p* = 0.035), possibly due to higher percentage of TN performed in the TALA group compared to the RP group (11:9). It is known that TN has a significantly higher risk of acute renal Injury in the short-term interval of 48 h postoperative and progression to chronic renal failure in the long-term interval of up to one year following surgery [11].

Considering the observed 30-day readmission rate of 15% in our cohort and comparing it to a recently published review paper by Kugreja et al. [12] that reported 30-day readmission rates of 4.2–6.1% for minimal invasive RN and 3.2–4.5% for PN, it must be acknowledged that (1) readmission rates were consistently understated by 17% to 29% across all major urological oncological surgeries [13] and (2) 83% (*n* = 5) of the readmission were in the TALA group, belonging to the initial section of the learning curve of a new established Access at our institution. We expect that this rate will significantly decrease considering the natural trajectory of the learning curve and increasing institutional caseload.

The 15% blood transfusion rate is also higher than expected and only concerned patients in the TALA group. This is a somewhat controversial finding as the estimated median blood loss is the same in both groups (175 mL). This finding could be related to a higher pre-operative hemoglobin level in the RP group, but this was not investigated in the current study. However, hemostatic agents could reduce transfusion requirements [14]. 

The 15% grade III-V complication rate suggests opportunities to improve safety outcomes. Additionally, the grade >2 complication rate of 6.47% was somewhat higher compared to a big reference French national data base [15]. Again, the numbers from our study should be contextualized, as they reflect the complication rate at the beginning of the learning curve, and it is well known that the rate of severe complications is higher in the initial phase before reaching the plateau phase [16]. 

Considering the specific complications, two patients in the RP group developed pneumothorax. There are several possible explanations for the development of pneumothorax during or following laparoscopic TN and PN. Known risk factors are the duration of surgery and the level of CO_2_ insufflation pressure, which causes higher CO2 absorption. It is debated whether the risk of pneumothorax development is higher during RP access, due to the lower compliance of the retroperitoneal cavity compared to the abdominal cavity or whether the smaller working space in the RP also promotes a higher risk of injury to the pleura or a combination of both factors [17]. In this context, an additional interesting point was reported by Bhardwaj et al., which investigated the anesthesiologic advantages of TP and RP access in children [18]. It was shown that RP access was associated with a higher CO_2_ absorption and increased pulmonary artery pressure which is of relevance in patients with cardiopulmonary pre-conditions [18]. In future studies, it would be desirable to investigate the anesthesiologic aspects of the TALA approach and thus determine if this approach also offers some advantage for patients with increased cardiopulmonary risks.

Concerning the case of duodenal perforation, the pathogenesis is difficult to interpret. It took place after a right-sided surgery, which could suggest an iatrogenic problem. However, iatrogenic bowel injuries usually occur within the first days after surgery, whereas the patient, who has a history of gastric ulceration developed the perforation after four weeks, which could also suggest a perforated ulcer.

The duration of the hospital stay is mainly based on our country’s health care system, which is based on DRG (diagnosis-related groups) and implements an ideal hospital stay from a financial point of view for the treating hospital (In the case of renal surgery about 6 days). 

The only publication of a similar approach, labeled as the “Trans-retro (TR) approach”, reports a case of a 42-year-old woman with a renal tumor near the hilus in the upper middle pole region of the kidney. Total operative time was 5 h and 19 min, with access complicated by multiple prior abdominal surgeries and extensive perirenal fat. Ischemia time was 20.5 min, and the patient was discharged on postoperative day three. The authors concluded, that TR is a promising option for posterior tumors, which may be an alternative for open surgery or laparoscopic surgery for surgeons unfamiliar with the RP access [7].

The validity of the study was limited, as data from the remaining eight patients were not evaluated in a systematic manner. Thus, to the best of our knowledge, this study is the first to report a small-scale but systematic comparison between the conventional RP approach and a modified hybrid TR approach.

In the literature, various advantages and disadvantages of the TP and the RP approaches are discussed. It has been shown that the pneumoperitoneum of the TP approach creates a larger working space in the abdomen resulting in a greater range of motion for the surgeon. At the same time, the access increases the risk of injury to intra-abdominal organs, primarily due to the necessary mobilization of the colon during nephrectomy [19]. In contrast, the RP approach poses a lower risk of injury but also a smaller working field due to the proximity of the trocars. 

Our study showed that TALA might have advantages compared to the RP approach. Improved visualization is one of the primary benefits; TALA potentially provides more direct sightlines to the renal hilum and surrounding structures by combining the initial capacious approach of transabdominal access with the targeted access of retroperitoneal dissection. This hybrid pathway can streamline the procedure and reduce the anatomical dissection complexities often encountered in pure RP approaches.

Additionally, the TALA’s strategic trocar placement is designed to mitigate surgical conflicts inherent to the constrained working space of the RP approach. TALA may offer a more ergonomic layout by leveraging the expanded space during the initial phase of the surgery for the placement of trocars, which is then followed by transposing the operative focus to a retroperitoneal working space, potentially reducing instrument clashing and improving maneuverability, which was not investigated in the present study. 

Our data on operative times, blood loss, and length of hospital stay suggest that the TALA approach could enhance patient outcomes. The median operation time was 207 min, the median blood loss was 175 mL, and the median hospital stay was 6 days, indicating a promising trend in surgical efficiency and patient recovery. While these findings are based on a limited series of the first 20 cases, they offer an encouraging benchmark for the potential benefits of TALA.

In contrast, the RP approach has been established for many years and the outcome corresponds to the plateau phase of the learning curve. 

Compared to the case of Regmi et al. [7], with an operation time of 5 h and 19 min, we can already achieve a shorter—although not significant—operation time with the TALA approach (207 min) compared to the RP approach (211 min). We therefore assume that with even more practice, we can further expand the advantages of the TALA approach over the RP approach. This is one of the strengths of our study. To our knowledge, we are the first to have investigated the TALA approach systematically and to have compared the data with the established RP approach. 

At first glance, the limitation of this approach could be upper pole tumors. However, we have been able to successfully operate on upper pole tumors with this approach. In such a case, sufficient mobilization of the kidney must be achieved so that the tumor can be sufficiently exposed. Theoretically, those located medially and directly proximal to the hilus could be limiting. We will try to investigate the extent to which localization is limiting in our follow-up study. 

The present study has several limitations that must be acknowledged. Firstly, the study design is inherently susceptible to selection bias, as the cases and controls were not randomly assigned. Secondly, the study is reliant on the accuracy and completeness of the medical records and data collection, as this information was not collected specifically for the study. Thirdly, the sample size of the study was relatively small, which limits the generalizability of the findings. Finally, the study was retrospective in nature, meaning that it was not possible to control for potential confounding variables. Furthermore, we strongly believe that trocar placement and instrument movement are much easier using TALA, leading to a better working space with more comfort for the surgeon. However, this question was not possible to answer in the retrospective setting. 

Considering these limitations and investigating the possible superiority of the TALA over the RP approach, we are currently conducting a prospective randomized trial (NCT05377632) to provide high-quality data to answer this question.

## 5. Conclusions

We report the first comparative study on TALA, which seems to be a safe and promising novel hybrid approach for renal surgery by combining the advantages of RP and TP. The information gained from this study provided the fundament for an ongoing randomized controlled trial.

## Figures and Tables

**Figure 1 cancers-16-00446-f001:**
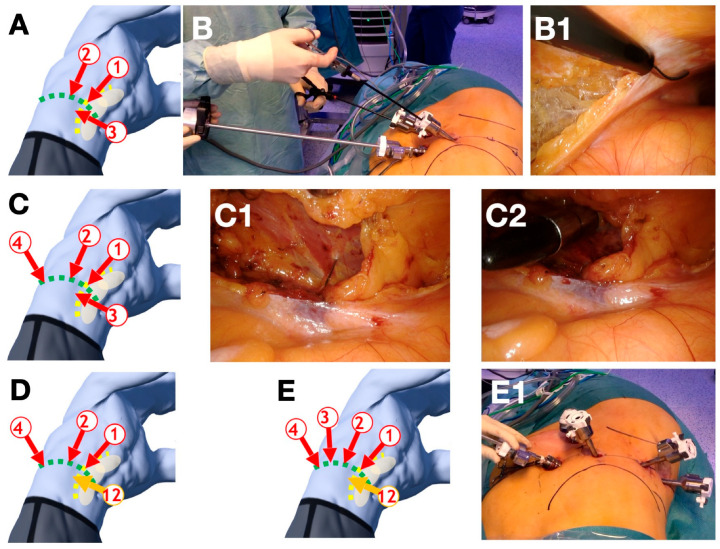
Step-by-step description of the trocar placement for the transabdominal lumbar approach (TALA). (**A**) The first trocar (1) is placed at the crossing of the lumbar line (green line) oriented in direction towards the umbilicus and the lateral boarder of the rectus abdominis muscle (yellow line). After placement of the first trocar (trocar 1) and establishing the pneumoperitoneum two additional DaVinci-trocars are placed (trocars 2 and 3). One (trocar 2) on the lumbar line at a distance of approximately 8 cm and a second DaVinci trocar (trocar 3). It is important to achieve a slight triangulation and a distance of 6–8 cm between the other trocars. (**B**) Next, the retroperitoneum is opened by a conventional laparoscopic incision of the Toldt’s line and blunt dissection (**B1**). (**C**) The lateral abdominal wall is exposed (**C1**) in the retroperitoneum, one additional trocar (trocar 4) is placed under direct vision (**C2**). (**D**) The DaVinci trocar 3 is replaced by a 12 mm assistant trocar (trocar 12). (**E**) The last DaVinci trocar (trocar 3) is now placed between (trocar 2 and trocar 4) with a distance of approximately 6–8 cm in between. (Images (**A**,**C**–**E**) courtesy of magicposer).

**Figure 2 cancers-16-00446-f002:**
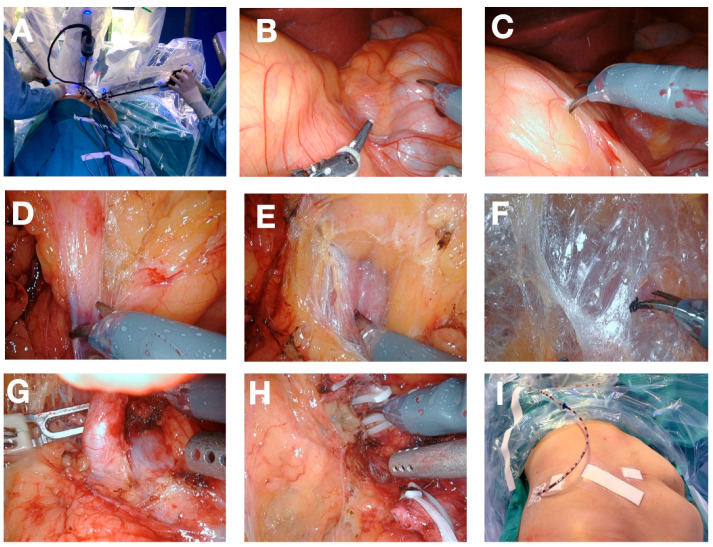
Description of the procedure showing the intraoperative view of the operation (**A**) docked robot; (**B**) view of the anatomy after docking: Colon ascendens (beneath right instrument), kidney shape left to left instrument; (**C**) incision of the peritoneum between kidney and colon; (**D**) dissection of the lower pole adherences; (**E**) view of psoas muscle; (**F**) avascular plane between psoas muscle and kidney (holding up kidney); (**G**) Renal artery, (**H**) dissection of the renal vein after dissection of the renal artery and clamping of the renal vein; (**I**) final postoperative view after undocking of robot and wound closure.

**Table 1 cancers-16-00446-t001:** Patient and Tumor Characteristics.

	RP	TALA	*p*-Value
Patient Characteristics			
Total no. Patients	20	20	
Median age (range)	65 (27–81)	67 (41–84)	0.62
Median BMI (range)	28.3 (22.1–34.3)	27.5 (17.8–35.9)	0.25
Sex (male:female), *n* (%)	12:8 (60:40)	14:6 (70:30)	0.74
ASA Score (1:2:3:4), %	5:70:25:0	5:50:45:0	0.23
Previous abdominal surgeries (Yes:No), *n* (%)	9:11 (45:55)	5:15 (25:75)	0.32
Type of surgery (Nephrectomy:Partial Nephrectomy), *n* (%)	7:13 (35:65)	9:11 (45:55)	0.75
Tumor Characteristics n (%)			
Benign	7 (35)	3 (15)	0.27
Malignancies-RCC	12 (60)	15 (75)	0.50
Clear cell	9	13	0.34
Papillary	2	1	1.00
Other	1	1	1.00
Non-functional kidney	1 (5)	2 (10)	1.00

**Table 2 cancers-16-00446-t002:** Intraoperative and Postoperative Characteristics, 30-Days Morbidity.

	RP	TALA	*p*-Value
Intraoperative Characteristics			
Median total operative time, min (range)	211 (154–289)	207 (124–306)	0.51
Median ischemia time, min (range)	15 (9–30)	17 (11–22)	0.61
Median estimated blood loss, mL (range)	175 (50–450)	175 (30–800)	0.98
R0-Resection, %	92	93	1.00
R1-Resection, %	8	7	1.00
Postoperative Characteristics			
Median hospitalization, d (ranges)	6 (4–9)	6 (4–8)	0.80
Median Analgesia surgery day (WHO level scheme), range	3 (1–4)	3 (1–4)	1.0
Median Analgesia POD 1 (WHO level scheme), range	2 (1–4)	1 (1–3)	0.53
Median Analgesia POD 2 (WHO level scheme), range	1 (1–3)	1 (1–3)	0.12
Median Analgesia POD 3(WHO level scheme), range	1 (0–3)	1 (0–3)	0.16
Median Analgesia POD 4(WHO level scheme), range	1 (0–3)	1 (0–3)	0.59
Median Analgesia POD 5(WHO level scheme), range	1 (0–3)	1 (0–3)	0.29
Median Change in Kidney function (µmol/L), range	−8 (−20–+75)	−29 (−24–+114)	0.02
Median Change in Hemoglobin (g/dL), range	−2.4 (0.1–6.9)	−2.5 (1.4–4.2)	0.60
30-Days Morbidity			
Complications (Clavien–Dindo), *n*			
No Complications/Grade I	17	15	0.44
Grade II	0	2	0.15
Grade III	3	2	0.64
Grade IV	0	1	0.32
Readmission, *n*			
Rehospitalization, *n*	1	5	0.05
IMC (intermediate care unit)	1	1	1.0
ICU (intensive care unit)	0	1	1.0
Blood transfusion, %			
Yes	0	15	0.23
No	100	85	0.23

## Data Availability

Data are contained within the article.

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
