# Peer review of "The Transabdominal Lumbar Approach (TALA) for Robotic Renal Surgery—A Retrospective Single-Center Comparative Study and Step-by-Step Description of a Novel Approach"

_cancers, 2024, doi:10.3390/cancers16020446_

Round 1
Reviewer 1 Report (Previous Reviewer 2)
Comments and Suggestions for Authors
The paper has been improved.
Author Response
Comment Reviever 1:
The paper has been improved.
Response Authors:
Many thanks for your feedback. We are happy that we were able to correctly answer your earlier suggestions and comments.

Reviewer 2 Report (New Reviewer)
Comments and Suggestions for Authors
The authors are to be commended for reporting an original and interesting technique and further analysing this in a planned randomised controlled trial.
My comments:
- Continuous variables are presented as mean (range): as this is a small patient series, values are not normally distributed and the median is a better parameter with range or interquartile range. For the same reason, T-test should not be used, only non-parametric tests.
- Was there Ethical Committee approval of the study ? Did patients consent for retrospective use of their data / images ?
- Surgical technique description:
*How is the patient positioned for TALA ? What robotic device was used (X or Xi) ?
*Please provide surgical description for RP surgery or add reference.
“The first trocar is placed at the intersection of the lumbar line and the lateral border of the rectus abdominis muscle via mini laparotomy access. “
Please make it clear for the readers how the lumbar line is defined exactly ? What anatomical landmarks are used to draw this line ?
- What was the Airseal intra-abdominal CO2 pressure ? This could be interesting because some patient developed pneumothorax postoperatively.
- How were patients selected to receive either RP or TALA in the analyzed patients ?
- Complications: one patient had a duodenal perforation. Was this a right sided surgery ? Was the duodenal perforation iatrogenic or did it develop later due to a perforated ulcer ?
Discussion:
- Our single-center data confirm safety and feasibility of this new hybrid technique and show non-inferiority compared to the standard retroperitoneal approach.
I agree on the safety and feasibility, but you cannot state non-inferiority. You have operated on 20 patients, there were some high-grade complications and 5 of them had a re-admission. Please rephrase.
- Hospital stay (6 days) is quite long as compared to other series (2-3 days). Please comment on the standard planned hospital stay following TN or PN in your institution/country.
- In the future: how would you select patients for one or the other approach ? Or would you perform TALA in all ? Do you feel this is feasible for upper pole tumors ?
- is this technique possible on other platforms as well do you think ? (Si ? SP ? Hugo RAS ? …)
Comments on the Quality of English LanguageSurgical technique and discussion: some wording errors
Author Response
Dear Editor
We would like to thank you and both reviewers for the kind and positive evaluation of our manuscript.
We have made a strong effort to ameliorate the manuscript according the comments and suggestions.
Reviewer 2
Comment 1
The authors are to be commended for reporting an original and interesting technique and further analysing this in a planned randomised controlled trial.
My comments:
- Continuous variables are presented as mean (range): as this is a small patient series, values are not normally distributed and the median is a better parameter with range or interquartile range. For the same reason, T-test should not be used, only non-parametric tests.
Response:
Many thanks for this good comment. We absolutely agree. We have performed new statistical analysis according to your suggestion and have changed the numbers in the tables and the corresponding section of the manuscript including changes in the statistics part of the method section. All changes are marked using the track-change mode.
Comment 2
Was there Ethical Committee approval of the study ? Did patients consent for retrospective use of their data / images ?
Response:
We have reiveved ethical approval for this study from the national ethics committee. All patients have signed a document for the use of data and images. In addition to the bottom line statement “Institutional Review Board Statement” we have now inserted some sentences in the methods section of the manuscript covering ethical considerations.
The following sentences were added in the methods section of the manuscript:
The study was conducted in accordance with the Declaration of Helsinki,and approved by the Cantonal Ethics Committee (Basec-No. 2023-00264). All patients have signed a document for the use of data and images.
Comment 3
Surgical technique description:
3.1) How is the patient positioned for TALA ? What robotic device was used (X or Xi) ?
3.2) Please provide surgical description for RP surgery or add reference.
3.3) The first trocar is placed at the intersection of the lumbar line and the lateral border of the rectus abdominis muscle via mini laparotomy access. “
Please make it clear for the readers how the lumbar line is defined exactly ? What anatomical landmarks are used to draw this line ?
Response:
Many thanks for these important comments. We agree that a more detailed description of the surgical technique can help the reader to better understand the technique. We have added more detailed descriptions of the technique by answering your comments.
3.1) Patients were positioned in a bent 70° flank position. We used a Xi da Vinci robotic system (Intuitive, USA).
3.2) A detailed description of the RP surgery was added to the surgical technique section of the manuscript.
3.3) We have added “dorsally between the iliac crest and the lowest rib and ventrally in the direction of the umbilicus”.
Comment 4
What was the Airseal intra-abdominal CO2 pressure? This could be interesting because some patient developed pneumothorax postoperatively.
Response:
That is a good point because actually the AirSeal-system allows to better operate also with lower pressures, which we recently have discussed within our team. However, for the present study we have used a constant pressure of 12mmHg in all cases. We have added this information in the text.
Comment 5
How were patients selected to receive either RP or TALA in the analyzed patients ?
Response:
After TALA was introduced in our institution, it was generally used more often. The decision for access for a specific patient ultimately remained a team decision.
We have added this statement in the methods section of the manuscript.
Comment 6
Complications: one patient had a duodenal perforation. Was this a right sided surgery ? Was the duodenal perforation iatrogenic or did it develop later due to a perforated ulcer ?
Response:
That is a very good question. In fact it was a right sided surgery, which could suggest that it could be a iatrogenic problem. However, iatrogenic injuries usually occur within the first days after surgery, whereas the patient, who has a history of gastric ulceration developed the perforation after four weeks, which could also suggest a perforated ulcer.
We have made a comment in the discussion section of the manuscript.
Comment 7
Discussion:
Our single-center data confirm safety and feasibility of this new hybrid technique and show non-inferiority compared to the standard retroperitoneal approach.
I agree on the safety and feasibility, but you cannot state non-inferiority. You have operated on 20 patients, there were some high-grade complications and 5 of them had a re-admission. Please rephrase.
Response:
Thank you for this comment. Of course we absolutely agree. We have removed the term non-inferiority form the text and have rephrased the sentence.
Comment 8
Hospital stay (6 days) is quite long as compared to other series (2-3 days). Please comment on the standard planned hospital stay following TN or PN in your institution/country.
Response:
Many thanks for this comment. Of course we agree that this point needs to be clarified. In fact this has more to do with our countries health care system which is based on DRG (Diagnosis Related Groups) and implements an ideal hospital stay in a financial point of view for the treating hospital. In the case of renal surgery it is about 6 days. We have added a comment in the discussion section of the manuscript.
Comment 9
- In the future: how would you select patients for one or the other approach ? Or would you perform TALA in all ? Do you feel this is feasible for upper pole tumors ?
Response:
As we have mentioned in the discussion, we are conducting a randomized controlled study at the moment, where we do not distinguish between upper and under pole tumours. We have successfully performed TALA on upper pole tumors. In these cases the kidney needs to be mobilized more to have a good exposure to the tumor. The only area which it might be very hard to reach is the medial upper pole, proximal to the hilus. This might be a limitation of the approach. We aim to answer these questions with our next study. However, we have now made a comment covering these thoughts in the present manuscript.
Comment 10
Is this technique possible on other platforms as well do you think ? (Si ? SP ? Hugo RAS ? …)
Response:
We believe that the idea of this technique could easily be adapted to other platforms. This is actually a very good comment. We have added a small sentence in the discussion part.
“Even though we only used the da vinci system in this study, we assume that the principle could also be adapted for other robotic platforms.”
Comment 11
Comments on the Quality of English Language: Surgical technique and discussion: some wording errors.
Response:
We have double-checked the complete manuscript again for wording errors and have made changes in the manuscript, which are highlighted in the track-change mode.

Round 2
Reviewer 2 Report (New Reviewer)
Comments and Suggestions for Authors
Thank you for addressing all comments.
Regarding the duodenal perforation: if it occurred 4 weeks after surgery, in my opinion it is very likely not related to the surgery as such, but due to duodenal ulcer.
Comments on the Quality of English Languageokay
This manuscript is a resubmission of an earlier submission. The following is a list of the peer review reports and author responses from that submission.
Round 1
Reviewer 1 Report
Comments and Suggestions for Authors
I read with interest the article by Franziska Maria Heining et al. entitled: "TALA: The transabdominal lumbar approach for robotic renal surgery"
The TALA technique described in the article is interesting, however the preliminary results of the study do not demonstrate any real benefit of this technique compared to the RP approach. Therefore, I do not consider this study to be particularly interesting at present.
Nonetheless, I would like to highlight a few further comments
• Table 1 and table 3: p-values are missing
• Table 1 and table 2 can be merged
• Tables 3-4-5 can be merged
• In the legend of table 5, IMC and IDIS must be defined
• In the discussion it is necessary to mention the increased costs that the TALA technique foresees as the costs of laparoscopy are associated with those of robotics (already high in itself).
• The results are substantially superimposable in the two groups. The conclusions of the abstract should be congruent
Author Response
We thank the reviewer very much for the good comments and also for finding our technique interesting. We agree that it has the potential to be a good alternative to the traditional approaches in the future. These are the first results in initial experiences with this technique. However, the manuscript already shows at this stage that the technique is comparably good. This is even at a stage where a certain learning curve undoubtedly plays a role. We are currently in the recruitment phase of a randomised trial comparing these two approaches. This will not only assess standard perioperative data (time, blood loss, etc.), but also clarity, comfort for the surgeon and instrument conflicts. However, we are convinced that it is important to publish the first patient data and thus the early learning curve now in order to be able to document the development of this technique later on.
We are happy to respond point by point to your very good comments.
Table 1 and 2: thank you very much for your comment. We have now added the missing p-values in the tables and combined the table into a single table. We have also combined tables 3-4-5 into a single table.
Please excuse us for not defining the ICU and IMC abbreviations. We have now provided the explanation into the table
Reviewer 2 Report
Comments and Suggestions for Authors
The authors propose a novel approach for robotic kidney surgery called TALA (Trans-Abdominal Lateral Approach), which combines the retroperitoneal and transperitoneal approaches.
Overall, the paper is interesting. However, here are a few remarks and suggestions for improvement:
1. Improve the quality of Figure 1: It is important to enhance the clarity of Figure 1, particularly by providing a clearer depiction of the port positions.
2. Expand the description of the surgical approach: The core focus of the paper should be on the detailed description of the surgical approach, including the specific port positions, patient positioning, and key steps involved in the renal surgery. The current description is too concise and should be expanded to provide a more comprehensive understanding of the technique.
3. Include intraoperative pictures: Adding intraoperative images, such as overviews of the operative field after docking and images illustrating the identification and dissection of the renal artery, would be helpful for readers to visualize the procedure better.
4. Standardize the evaluation of surgical experience: The authors mention that this approach allows for a less "conflicting" surgery. It would be beneficial to consider the possibility of standardizing this assessment by using a Likert scale or similar methodology to quantify the perceived reduction in conflicts. This would provide a more objective evaluation of the approach.
5. Address the observed readmission rate, blood transfusions, and complications: The paper states a 20% readmission rate, 15% blood transfusion rate, and 20% grade >2 complication rate. It is important to provide a thorough discussion defending these outcomes, explaining the factors that may have contributed to them, and considering any potential strategies for reducing these rates in the future.
6. Reframe the conclusion about TALA's superiority: The statement that "TALA might be superior to the RP approach due to the better intraoperative overview with fewer conflicts of instruments and a resulting increased control for the surgeon in the operative field" is not supported by the data presented. It would be advisable to either remove or soften this statement to align with the findings of the study.
7. Highlight the advantages of TALA: Since TALA and the RP approach appear to be fully comparable, it is essential to provide a scientific, data-supported description of the advantages offered by TALA. This could include discussing potential benefits such as improved visualization, reduced surgical conflicts, shorter operative times, or enhanced patient outcomes.
Comments on the Quality of English Language
Good
Author Response
Thank you very much for your kind assessment of our manuscript and at the same time your detailed and helpful comments. We have now invested a lot of time in improving the manuscript according to your comments and are convinced that it has thus also gained in quality thanks to you. We are happy to address your comments point by point:
- Many thanks for your suggestion. We have now improved the quality of figure 1, especially regarding the port placement.
- Indeed, we also believe that the detailed description of the approach is one of the main points of this paper. Therefore, we have expanded this section in the manuscript. The changes have been highlighted by track changes in the “surgical technique” section of the manuscript (page 3).
- Thank you for this suggestion. We have now added further pictures from the intraoperative field, especially after docking and from the hilus/artery.
- That is a very good point, thank you for that. We have now set up a prospective randomized trial where we have implemented a Likert-like questionnaire for the console-surgeon as well as for the table-side assistant. We are convinced that this will allow more insight. We are planning to publish these data in the future. However, the present manuscript represents an early series of this approach. Unfortunately we are not able to provide this data in a retrospective manner.
- We thank the reviewer for addressing this point. We have now written a more detailed explanation in the discussion section of the manuscript, which we have highlighted in track changes.
- You are absolutely right with this point. Even though we believe that this is the case, the statement is not supported by the data at the moment. We have now softened the statement: “Our study showed that TALA might have advantages comparedto the RP approach due to the better intra-operative overview with fewer conflicts of instruments and a resulting increased control for the surgeon in the operative field, which was not investigated in the present study.
That is a very helpful suggestion. We have now implemented a section to more highlight the advantages of TALA. The sentences are marked with track changes in the discussion section of the manuscript.